# Strongly exchange-coupled and surface-state-modulated magnetization dynamics in $Bi_2Se_3$/ yttrium iron garnet heterostructures

Y.T. Fanchiang[1], K.H.M. Chen[2], C.C. Tseng[2], C.C. Chen[2], C.K. Cheng[1], S.R. Yang[2], C.N. Wu[2], S.F. Lee[3], M. Hong[1] & J. Kwo[2]

Harnessing the spin–momentum locking of topological surface states in conjunction with magnetic materials is the first step to realize novel topological insulator-based devices. Here, we report strong interfacial coupling in $Bi_2Se_3$/yttrium iron garnet (YIG) bilayers manifested as large interfacial in-plane magnetic anisotropy (IMA) and enhancement of damping probed by ferromagnetic resonance. The interfacial IMA and damping enhancement reaches a maximum when the $Bi_2Se_3$ film approaches its two-dimensional limit, indicating that topological surface states play an important role in the magnetization dynamics of YIG. Temperature-dependent ferromagnetic resonance of $Bi_2Se_3$/YIG reveals signatures of the magnetic proximity effect of $T_C$ as high as 180 K, an emerging low-temperature perpendicular magnetic anisotropy competing the high-temperature IMA, and an increasing exchange effective field of YIG steadily increasing toward low temperature. Our study sheds light on the effects of topological insulators on magnetization dynamics, essential for the development of topological insulator-based spintronic devices.

[1] Department of Physics, National Taiwan University, Taipei 10617, Taiwan. [2] Department of Physics, National Tsing Hua University, Hsinchu 30013, Taiwan. [3] Institute of Physics, Academia Sinica, Taipei 11529, Taiwan. Correspondence and requests for materials should be addressed to S.F.L. (email: leesf@gate.sinica.edu.tw) or to M.H. (email: mhong@phys.ntu.edu.tw) or to J.K. (email: raynien@phys.nthu.edu.tw)

The development of spintronics relies crucially on control of spin-polarized currents, which carry spin angular momenta that can be utilized to manipulate magnetic moments through spin-transfer processes. Spin currents can be generated by the spin Hall effect[1] in a heavy metal, or by exploiting the spin structure of some two-dimensional (2D) electron systems. A promising candidate of such a 2D system is the surface state of topological insulators (TIs). TIs are emergent quantum materials hosting topologically protected surface states, with dissipationless transport prohibiting backscattering[2,3]. Strong spin–orbit coupling (SOC) along with time reversal symmetry (TRS) ensures that the electrons in the topological surface states (TSSs) have their direction of motion and spin locked to each other[2,4,5]. The spin–momentum locking permits efficient interconversion between spin and charge currents. To date, several methods have been adopted to estimate the spin-charge conversion efficiency of TIs, either by using microwave-excited dynamical method[6–10] (e.g., spin pumping and spin–torque ferromagnetic resonance (ST-FMR)) or thermally induced spin injection[11]. Very large values of spin-charge conversion ratio have been reported[7,9,10]. Recently, TIs are shown to be excellent sources of spin–orbit torques (SOT) for efficient magnetization switching[12].

When a TI is interfaced with a magnetic layer, the interfacial exchange coupling can induce magnetic order in TIs by the magnetic proximity effect (MPE) and break the TRS[13–16]. The resulting gap opening of the Dirac state is necessary to realize novel phenomena such as topological magneto-electric effect[17] and quantum anomalous Hall effect[18,19]. Since the MPE and spin-transfer process rely on interfacial exchange coupling of TI/ferromagnet, understanding the magnetism at the interface has attracted strong interests in recent years. Several techniques have been adopted to investigate the interfacial static magnetic properties, including spin-polarized neutron reflectivity[15,20], second harmonic generation[21], electrical transport[14,22], and magneto-optical Kerr effect[14]. All these studies clearly indicate

the existence of MPE resulting from exchange coupling and strong SOC in TIs. Specifically, a room-temperature magnetic order induced by MPE in $EuS/Bi_2Se_3$ has been reported recently[15]. Through exchange coupling between the TSS and EuS layer, the induced magnetic moments exhibited perpendicular magnetic anisotropy (PMA) that can potentially open a gap of TSS. For TI/yttrium iron garnet (YIG) bilayer, however, the interfacial magnetic anisotropy and the resulting magnetization dynamics under the influence of TSS are still largely unknown. It is equally important to understand how the interfacial exchange coupling affects the magnetization dynamics of $Bi_2Se_3$/YIG because of the wide applications of YIG. For example, TIs can enhance the magnetic anisotropy, introduce additional magnetic damping, and greatly alter the dynamical properties of the ferromagnetic layer, as commonly observed in ferromagnet/heavy metals systems[23–25]. The enhanced damping is visualized as larger linewidth of FMR spectra[23–25]. Given the volatile surface band structure depending on the TI thickness[26], the adjacent materials[27], and the magnetism at the interfaces[28], experimental study on how the magnetization dynamically responds to the TSS is still lacking, which is a topic not only important for spintronics but also fundamental for physics.

In this work, we have investigated the magnetization dynamics via FMR in ferrimagnetic insulator YIG under the influence of the prototypical three-dimensional (3D) TI $Bi_2Se_3$[29]. We choose YIG as the ferromagnetic layer because of its technological importance, with high $T_C$ ~550 K and extremely low damping coefficient $\alpha$[30]. When YIG is interfaced with TIs, its good thermal stability minimizes the interdiffusion of materials. Through the $Bi_2Se_3$ thickness dependence study, we observed a strong modulation of FMR properties attributed to the TSS of $Bi_2Se_3$. The temperature-dependent study unraveled an effective field parallel to the magnetization direction existing in $Bi_2Se_3$/YIG. Such an effective field built up as the temperature decreased, which was utilized to demonstrate the zero-applied-field FMR of YIG. Furthermore, we identified a possible signature of MPE of

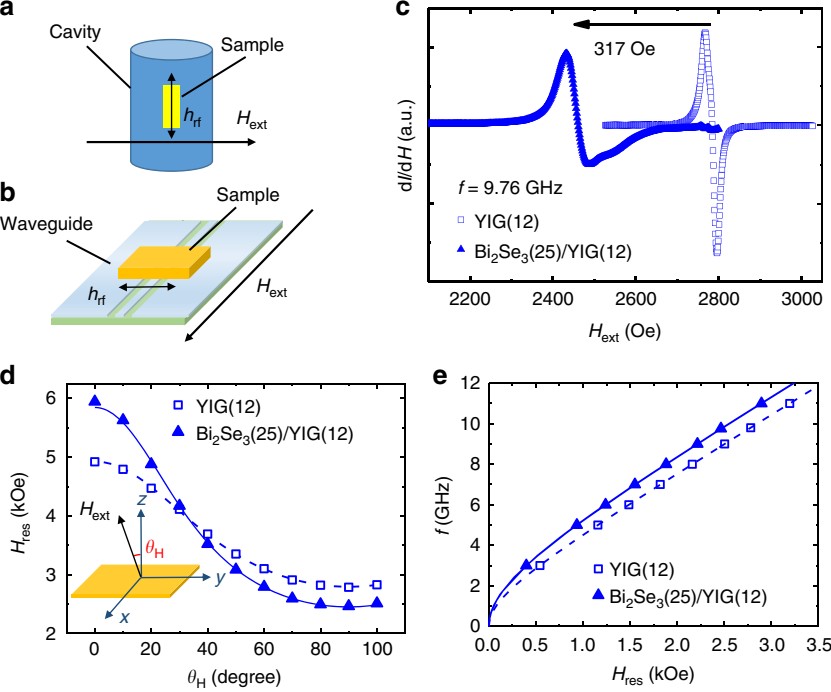

**Fig. 1** Schematic diagrams and results of the angle- and frequency-dependent FMR measurements. **a**, **b** FMR using the cavity and co-planar waveguide configuration for angle- and frequency-dependent study, respectively. A dc external field $H_{ext}$ was applied and $h_{rf}$ denotes the microwave field. **c** FMR spectra of $Bi_2Se_3(25)/YIG(12)$ and YIG(12) measured by the cavity. **d**, **e** $\theta_H$ and $f$ dependence of $H_{res}$ of $Bi_2Se_3(25)/YIG(12)$ and YIG(12), respectively

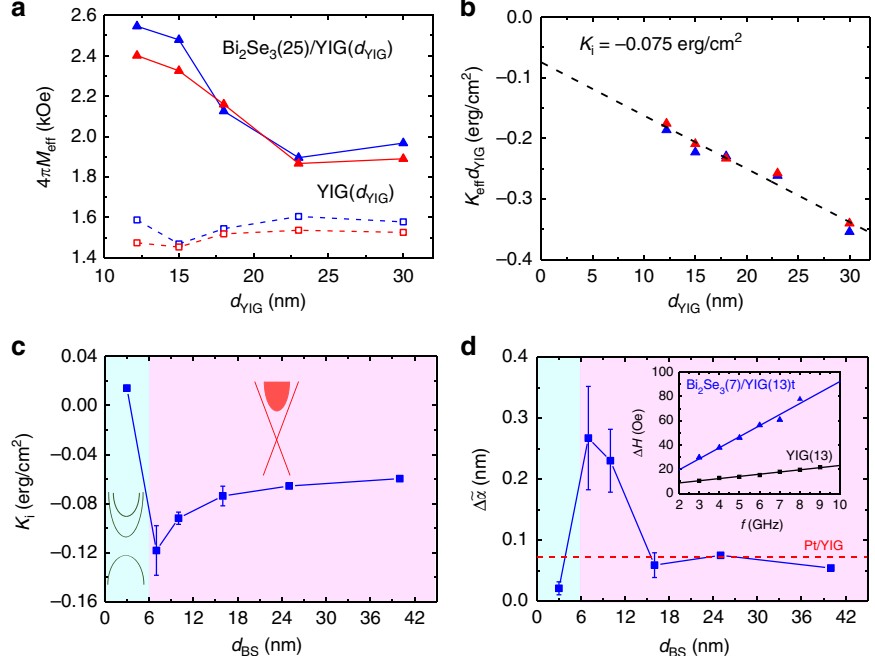

**Fig. 2** YIG and $Bi_2Se_3$ thickness dependence of FMR characteristics of $Bi_2Se_3$/YIG. **a** The $d_{YIG}$ dependence of $4\pi M_{eff}$ of $Bi_2Se_3$/YIG (solid triangles) and YIG (hollow squares) obtained from $\theta_H$ (red) and $f$ (blue) dependent FMR. **b** The $K_{eff}d_{YIG}$ vs. $d_{YIG}$ plot for determining $K_i$ using a linear fit. The intercept of the $y$-axis corresponds to the $K_i$ value. **c** $d_{BS}$ dependence of $K_i$. The figure is divided into two regions. For $d_{BS} > 6$ nm, the Dirac cone of TSS is intact, with the Fermi level located in the bulk conduction band. For $d_{BS} < 6$ nm, a gap and quantum well states form. **d** The $d_{BS}$ dependence of normalized damping enhancement $\Delta\tilde{\alpha}$. The inset shows $\Delta H$ as a function of $f$ $Bi_2Se_3(7)$/YIG(13) and YIG(13) for calculating $\alpha_{BS/YIG} - \alpha_{YIG}$. The red dashed line shows the typical value of $\Delta\tilde{\alpha}$ of Pt/YIG for comparison. The error bars indicate standard deviations of at least four samples

$T_C$ as high as 180 K manifested as enhanced spin pumping in a fluctuating spin system, as well as a small emerging PMA at low temperature in competition with in-plane magnetic anisotropy (IMA) extending to high temperature.

## Results

**Interfacial IMA in $Bi_2Se_3$/YIG.** The room-temperature FMR measurements were performed using a microwave cavity of frequency 9.76 GHz (Fig. 1a) and a broadband coplanar waveguide (Fig. 1b). The FMR spectra in Fig. 1c are compared for single layer YIG(12) and $Bi_2Se_3(25)$/YIG(12) bilayer (digits denote thickness in nanometer), showing a large shift of resonance field ($H_{res}$) ~317 Oe after the $Bi_2Se_3$ growth plus a markedly broadened peak-to-peak width $\Delta H$ for $Bi_2Se_3$/YIG. Figure 1d shows $H_{res}$ vs. applied field angle with respect to the surface normal $\theta_H$ for YIG (12) and $Bi_2Se_3(25)$/YIG(12). Larger variation of $H_{res}$ with $\theta_H$ in the bilayer sample was observed. When the applied field was directed in the film plane, clear negative $H_{res}$ shifts induced by $Bi_2Se_3$ were observed at all microwave frequencies $f$ as shown in Fig. 1e. The data in Fig. 1d, e can be fitted in the scheme of magnetic thin films having uniaxial PMA, the strength of which is characterized by the effective demagnetization field $4\pi M_{eff} = 4\pi M_s - H_{an} - H_{int}$, where $4\pi M_s$, $H_{an}$, and $H_{int}$ are the demagnetization field, the magnetocrystalline anisotropy field of YIG, and the interfacial anisotropy field induced by $Bi_2Se_3$, respectively. The fitting result shows an ~60% enhancement of $4\pi M_{eff}$ for the $Bi_2Se_3(25)$/YIG(12) bilayer sample. The large enhancement cannot be accounted for by an increase in the saturation magnetization $M_s$, which should amount to an additional magnetization of ~100 $\mu_B/nm^2$ for this sample. The MPE, even if it persists up to room temperature, is unlikely to induce the large amounts of magnetic moments. Furthermore, since the x-ray diffraction results in Supplementary Fig. 3(d) and (e) show that the YIG films did not gain additional strain after growing $Bi_2Se_3$, the

enhanced anisotropy cannot result from the change of magnetocrystalline anisotropy. We thus attribute the change of anisotropy mostly to the $H_{int}$. Based on the above discussion, we obtain $H_{int} = -926$ and $-1005$ Oe from Fig. 1d, e, respectively (see Supplementary Note 2). The minus sign indicates the additional anisotropy points in the film plane.

The above observations suggested the presence of interfacial IMA in $Bi_2Se_3$/YIG. To verify this, we systematically varied the thickness of YIG, $d_{YIG}$, while fixing the thickness of $Bi_2Se_3$. Figure 2a presents the $d_{YIG}$ dependence of $4\pi M_{eff}$ for single and bilayer samples. The $4\pi M_{eff}$ of single layer YIG was independent of $d_{YIG}$ varying from 12 to 30 nm. In sharp contrast, $4\pi M_{eff}$ of $Bi_2Se_3$/YIG became significantly larger, especially at thinner YIG, which is a feature of an interfacial effect. The $f$- and $\theta_H$-dependent FMR were performed independently to doubly confirm the trends. The interfacial IMA can be further characterized by defining the effective anisotropy constant $K_{eff} = (1/2)4\pi M_{eff}M_s = (1/2)(4\pi M_s - H_{an})M_s - K_i/d_{YIG}$, with the interfacial anisotropy constant $K_i = M_s H_{int}d_{YIG}/2$. The $K_{eff}d_{YIG}$ vs. $d_{YIG}$ data in Fig. 2b are well fitted by a linear function, indicating that the $d_{YIG}$ dependence presented in Fig. 2a is suitably described by the current form of $K_{eff}$. The intercept obtained by extrapolating the linear function corresponds to $K_i = -0.075$ erg/cm$^2$.

**TSS-modulated magnetization dynamics in $Bi_2Se_3$/YIG.** To further investigate the physical origin of the IMA, we next varied the thickness of $Bi_2Se_3$ ($d_{BS}$) to see how $K_i$ evolved with $d_{BS}$. Figure 2c shows the $d_{BS}$ dependence of $K_i$. Starting from the $d_{BS} = 40$ nm sample, the magnitude of $K_i$ went up as $d_{BS}$ decreased. An extremum of $K_i$ $-0.12 \pm 0.02$ erg/cm$^{-2}$ was reached at $d_{BS} = 7$ nm. An abrupt upturn of $K_i$ occurred in the region 3 nm $< d_{BS} < 7$ nm. The $K_i$ magnitude dropped drastically and exhibited a sign change in the interval. Furthermore, the $K_i$ value of 0.014 erg/cm$^2$ at $d_{BS} = 3$ nm corresponds to weak interfacial PMA.

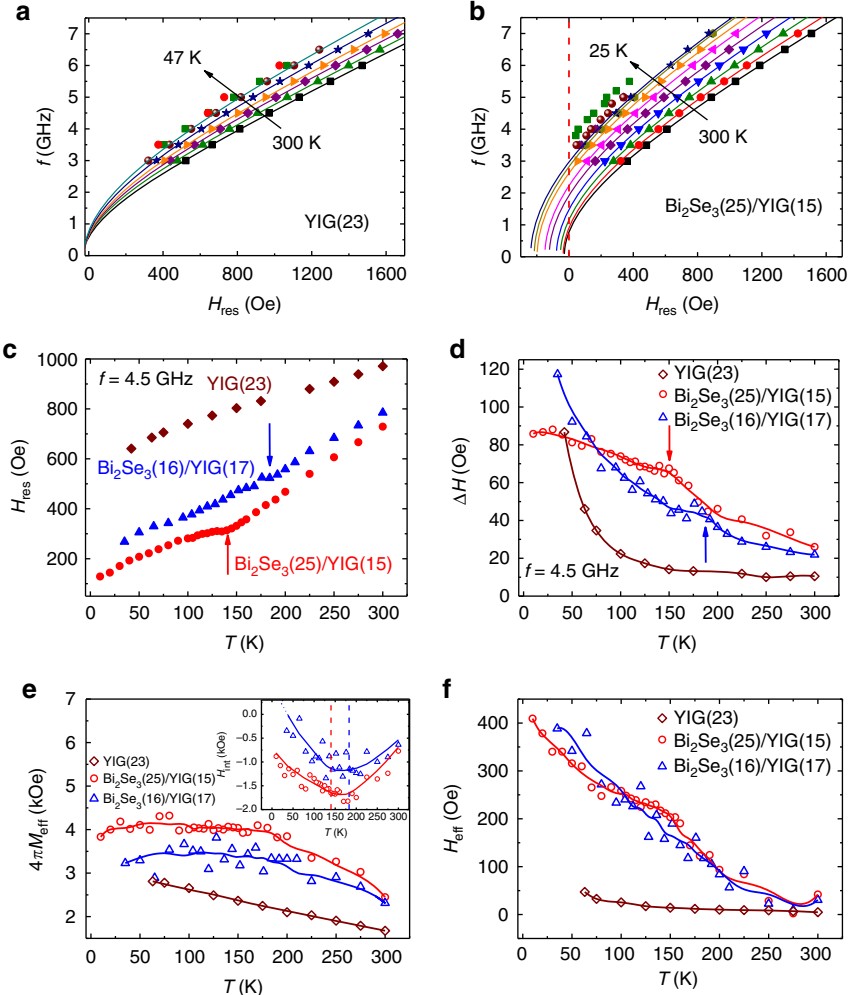

**Fig. 3** Temperature dependence of FMR characteristics of $Bi_2Se_3$/YIG. **a**, **b** $f$ vs. $H_{res}$ data for various $T$ for YIG(23) and $Bi_2Se_3$(25)/YIG(15), respectively. Solids lines are fitted curves using Eq. (2). **c**, **d**, **e**, and **f** $T$ dependence of $H_{res}$, $\Delta H$, $4\pi M_{eff}$, and $H_{eff}$ of one YIG(23) single layer and two $Bi_2Se_3$/YIG bilayer samples, $Bi_2Se_3$(25)/YIG(15) and $Bi_2Se_3$(16)/YIG(17), respectively. The arrows in **c** and **d** denote the position of the hump-like features. Solid lines are guides of the eyes obtained by properly smoothing the experimental data. The inset of **e** shows the $T$ dependence of $H_{int}$. The dashed lines indicate the hump position shown in **c** and **d**

The sizable interfacial IMA can be expected given the large SOC of $Bi_2Se_3$. One possible mechanism is that the electrons at the interface re-distribute upon hybridization between the Fe $d$-orbital of YIG and the Dirac surface state of $Bi_2Se_3$. Recent theoretical study on $EuS$/$Bi_2Se_3$ bilayers indicates that in addition to the strong SOC, TSS play a crucial role in mediating the exchange coupling of the ions in the magnetic layer[31]. The hybridization between TSS and the magnetic layer can overall enhance the magnetic anisotropy energy that is inherent at the interface[31]. Although in general an interfacial magnetic anisotropy may not necessarily be related to the topological nature of materials, here we attribute the interfacial magnetic anisotropy of $Bi_2Se_3$/YIG to the TSS based on the unique $d_{BS}$ dependence of $K_i$, that cannot be accounted for by the strain or chemical mixing effects. Note that possible interdiffusion of materials at the interface can also lead to an interfacial magnetic anisotropy. As shown in Supplementary Fig. 3(c), the transmission electron microscope (TEM) image reveals an ~1 nm interfacial layer. However, the interdiffusion is unlikely to play a dominant role in the interfacial magnetic anisotropy since $K_i$ varied significantly with $d_{BS}$ up to 40 nm, and cannot account for the modulated dependence of $K_i$ with $d_{BS}$, especially under 20 nm. We now consider how the $Bi_2Se_3$ band structure evolves with $d_{BS}$. Based on

previous investigation on surface band structure of ultrathin $Bi_2Se_3$[26], $d_{BS} = 6$ nm was identified as the 2D quantum tunneling limit of $Bi_2Se_3$. When $d_{BS} < 6$ nm, the hybridization of top and bottom TSS developed a gap in the surface states. Spin-resolved photoemission study later showed that the TSS in this 2D regime exhibited decreased in-plane spin polarization[32]. The modulated spin texture may lead to the weaker interfacial magnetic anisotropy than that in the 3D regime[32,33]. We thus divide Fig. 2c into two regions and correlate the systematic magnetic properties with the surface state band structure. The sharp change of $K_i$ around $d_{BS} < 6$ nm strongly suggests that the interfacial IMA in $Bi_2Se_3$/YIG is of topological origin.

The $\Delta H$ broadening in FMR spectra after growing $Bi_2Se_3$ on YIG indicates that $Bi_2Se_3$ introduced additional damping in YIG. Within the macrospin approximation, the damping enhancement can be normalized with respect to $d_{YIG}$ by defining $\Delta\tilde{\alpha} = d_{YIG}(\alpha_{BS/YIG} - \alpha_{YIG})$, where $\alpha_{BS/YIG}$ and $\alpha_{YIG}$ are the damping coefficient of $Bi_2Se_3$/YIG and YIG, respectively. Figure 2d displays the $d_{BS}$ dependence of $\Delta\tilde{\alpha}$. Similar to $K_i$ in Fig. 2c, $\Delta\tilde{\alpha}$ increased as $d_{BS}$ decreased, reached its maximum at $d_{BS} = 7$ nm with a very large value of ~0.27 nm, and then dropped abruptly in the interval of 3 nm $< d_{BS} < 7$ nm. For comparison, typical $\Delta\tilde{\alpha}$ of Pt/YIG, in which efficient spin pumping giving rise

to sizable $\Delta\tilde{\alpha}$[24], is indicated by the red dashed line. The inset shows $\Delta H$ vs. $f$ data for $Bi_2Se_3$ (7)/YIG(13) and YIG(13) fitted by linear functions. One can clearly see a significant change of slope, from which we determined $\alpha_{BS/YIG} - \alpha_{YIG}$ to be 0.014. In general, the large damping enhancement can have multiple origins, including spin-pumping effect, interlayer exchange coupling with other magnetic layers, and chemical reactions at the interface. However, the damping arising from the static exchange coupling from the MPE or any antiferromagnetic order at the interface is not expected at room temperature. Moreover, as previously mentioned, the slight interdiffusion at the interface is unlikely to be the major root cause of $\Delta\tilde{\alpha}$ varying over the wide range of $d_{BS}$. Instead, considering the closed $d_{BS}$ dependence of $K_i$ and $\Delta\tilde{\alpha}$, it can be seen that the trend of $\Delta\tilde{\alpha}$ in Fig. 2d stemmed from the strong coupling between TSS of $Bi_2Se_3$ and YIG—that is, the surface state band structure of $Bi_2Se_3$ profoundly affected the damping of YIG[6]. Through dynamical exchange, spin angular momenta were transferred from YIG to the TSS via the spin-pumping effect. The spin-pumping efficiency of an interface can be evaluated by the real part of spin mixing conductance $g_{\uparrow\downarrow}$ using the following relation[25]:

$$g_{\uparrow\downarrow} = \frac{4\pi M_s d_{YIG}}{g\mu_B}\left(\alpha_{BS/YIG} - \alpha_{YIG}\right) = \frac{4\pi M_s \Delta\tilde{\alpha}}{g\mu_B}, \qquad (1)$$

where $g$ and $\mu_B$, are the Landé $g$ factor and Bohr magneton, respectively. The maximum $g_{\uparrow\downarrow}$ value ($d_{BS} = 7$ nm) is calculated to be $\sim 2.2 \times 10^{15}$ cm$^{-2}$, about three times larger than that of a typical Pt/YIG sample. The large $g_{\uparrow\downarrow}$ of $Bi_2Se_3$/YIG implies an efficient spin pumping to an excellent spin sink of $Bi_2Se_3$. Note that the trend in Fig. 2d is distinct from that of the normal metal (NM)/ferromagnetic metal (FM) structures. In NM/FM, the $g_{\uparrow\downarrow}$ increases with increasing NM thickness as a result of vanishing spin backflow in thicker NM[34]. It is worth noting that the conducting bulk of $Bi_2Se_3$ can dissipate the spin-pumping-induced spin accumulation at the interface[6,35]. In this regard, the $d_{BS} = 7$ nm sample has the largest weight of surface state contribution to $g_{\uparrow\downarrow}$. Such unconventional $d_{BS}$ dependence of $g_{\uparrow\downarrow}$ implies that TSS plays a dominant role in the damping enhancement.

**Spin-pumping signature of MPE and observation of the exchange effective field.** Since the effects of TSS are expected to enhance at low temperature, we next performed temperature-dependent FMR on $Bi_2Se_3$/YIG. Two bilayer samples $Bi_2Se_3$(25)/YIG(15) and $Bi_2Se_3$(16)/YIG(17), and a single layer YIG(23) were measured for comparison. Figure 3a, b shows the $H_{res}$ vs. $f$ data at various temperatures $T$ for YIG(23) and $Bi_2Se_3$(25)/YIG(15). The $H_{res}$ of both samples shows negative shifts at all $f$ with decreasing $T$. The data of YIG(23) can be reproduced by the Kittel equation with increasing $M_s$ of YIG at low $T$. In sharp contrast, $Bi_2Se_3$(25)/YIG(15) exhibited negative intercepts at $H_{res}$, and the intercepts gained their magnitude when the sample was cooled down. This behavior of non-zero intercept is common for all of our $Bi_2Se_3$/YIG samples. Note that the Kittel equation in its original form cannot produce an intercept. To account for the behavior, a phenomenological effective field $H_{eff}$ is added to the Kittel equation, i.e.,

$$f = \frac{\gamma}{2\pi}\sqrt{(H_{res} + H_{eff})(H_{res} + H_{eff} + 4\pi M_{eff})}. \qquad (2)$$

The solid lines in Fig. 3b generated by the modified Kittel equation fitted the experimental data very well.

Figure 3c, d presents the $T$ dependence of $H_{res}$ and $\Delta H$ for the YIG(23) and two $Bi_2Se_3$/YIG samples. As we lowered $T$, all of the samples had decreasing $H_{res}$, which was viewed as the effect of the

concurrently increasing $M_{eff}$ and $H_{eff}$ as seen in Fig. 3a, b. On the other hand, $\Delta H$ built up with decreasing $T$. We first examined $\Delta H$ of the YIG(23) single layer. The $\Delta H$ remained relatively unchanged with $T$ decreasing from room temperature, and dramatically increased below 100 K. The pronounced $T$ dependence of $\Delta H$ or $\alpha$ has been explored in various rare-earth iron garnet and was explained by the slow-relaxation process via rare-earth elements or $Fe^{2+}$ impurities triggered at low $T$[36]. For sputtered YIG films, specifically, the increase in $\Delta H$ was less prominent in thicker YIG, indicating that the dominant impurities were located near the YIG surface[37]. Distinct from that of YIG(23), the $\Delta H$ progressively increased for the bilayer samples. We were not able to detect FMR signals with $\Delta H$ beyond 100 Oe due to the limited sensitivity of our co-planar waveguide. However, one can clearly see that, for $Bi_2Se_3$(25)/YIG(15) and $Bi_2Se_3$(16)/YIG(17), $\Delta H$ broadened owing to increased spin pumping at first. For $Bi_2Se_3$(25)/YIG(15), the $\Delta H$ curve gradually leveled off, and intersected with that of YIG(23) at $T \sim 40$ K. The seemingly antidamping by $Bi_2Se_3$ at low $T$ may be related to the modification of the YIG surface chemistry during the $Bi_2Se_3$ deposition. Additional analyses are needed to verify the scenario, which is, however, beyond the scope of this work. For the $Bi_2Se_3$(10)/YIG and $Bi_2Se_3$(7)/YIG samples, the damping had increased to such large magnitude below 150 K, and FMR could not be easily detected.

In both $H_{res}$ and $\Delta H$ curves, hump-like features located at $T = 140$ and 180 K (indicated by the arrows) were revealed for $Bi_2Se_3$(25)/YIG(15) and $Bi_2Se_3$(16)/YIG(17), respectively. We note that the humps are reminiscent of spin pumping into a fluctuating magnet close to its magnetic ordering temperature. As pointed out by Ohnuma et al.[38], the spin-pumping efficiency is governed by the momentum sum of imaginary part of dynamical transverse spin susceptibility $\chi_k^R$ of the spin sink:

$$g_{\uparrow\downarrow} \propto \sum_k \frac{1}{\omega_{rf}}\mathrm{Im}\chi_k^R(\omega_{rf}), \qquad (3)$$

where $k$ is the wave vector and $\omega_{rf}$ is the microwave angular frequency. For a ferromagnet, the $\chi_k^R$ is known to be divergent near its $T_C$[39]. Therefore, an enhancement of spin pumping is expected as the spin sink is close to its magnetic phase transition point[38,40,41]. In our system, a possibly newly formed magnetic phase would be the interfacial magnetization driven by the proximity effect, namely, $T_C = 140$ and 180 K for our $Bi_2Se_3$(25)/YIG(15) and $Bi_2Se_3$(16)/YIG(17), respectively. In fact, the $T_C$ values of our samples are in good agreement with the reported $T_C$ of 130 and 150 K in TI/YIG systems[14,22].

Using Eq. (2), we further determine the $T$ dependence of $4\pi M_{eff}$ and $H_{eff}$ of YIG(23) and the two bilayers samples, as shown in Fig. 3e, f. The $4\pi M_{eff}$ of YIG(23) became larger monotonically as previously discussed, while the $4\pi M_{eff}$ of the bilayer samples increased before reaching a maximum when $T$ was around 150 K, and then decreased slightly at low $T$. We further calculate the interfacial anisotropy field $H_{int}$ using $4\pi M_{eff}^{BS/YIG} - 4\pi M_{eff}^{YIG} \approx -H_{int}$. The inset of Fig. 3e shows the $T$ dependence of $H_{int}$. The magnitude of $H_{int}$ increased as the samples cooled down from room temperature at first. Upon crossing the temperature regions where the hump-like features are located, $H_{int}$ magnitude started to decrease with further decreasing $T$. Although the samples exhibit interfacial IMA ($H_{int} < 0$) within the temperature range of our measurement, further extending the trend of $Bi_2Se_3$(16)/YIG(17), specifically, leads to interfacial PMA ($H_{int} > 0$) below 40 K. The turning of $H_{int}$ curves around 150 K implied that a competing magnetic anisotropy was emerging, which favored perpendicular direction and effectively diminished the IMA that persisted up to room temperature.

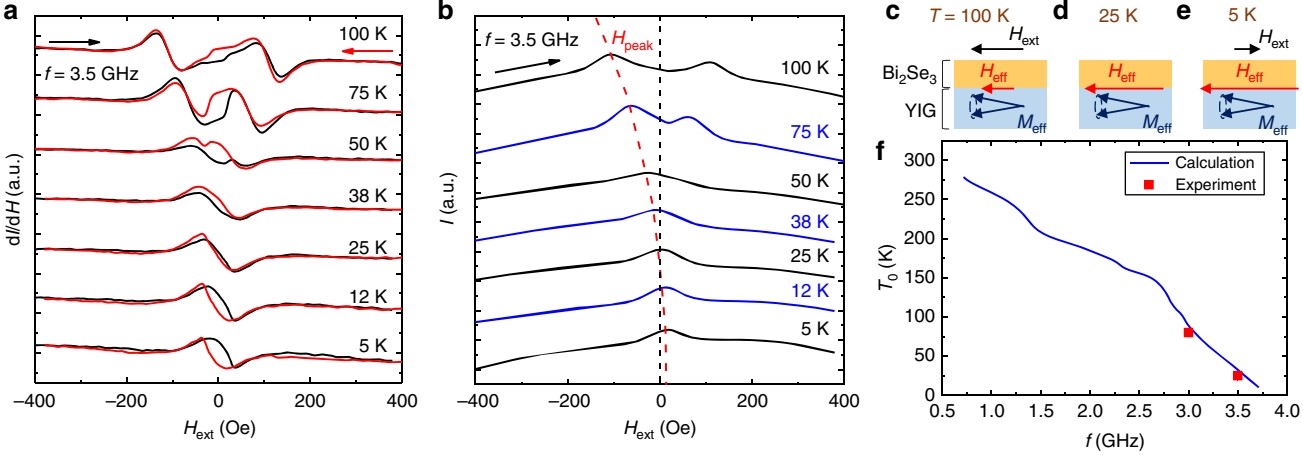

**Fig. 4** Zero-field FMR of the Bi$_2$Se$_3$(25)/YIG(15) sample. **a, b** FMR first derivative and microwave absorption spectra for various $T$, respectively. The arrows indicate the $H_{ext}$ sweep direction. The dashed line in **b** traces the $T$ evolution of the absorption peak $H_{peak}$. **c, d**, and **e** Schematics of the Bi$_2$Se$_3$/YIG sample when $T = 100$, 25, and 5 K, respectively. **f** Zero-field FMR temperature $T_0$ as a function of $f$

Observing that the turning of $H_{int}$ curves were in the vicinity of the individual hump temperature, we thus attribute the interfacial PMA to MPE in Bi$_2$Se$_3$/YIG. Our scenario is further supported by a theoretical model that considers the direct exchange coupling of TSS and an adjacent magnetic layer[31,42]. In this model, the calculated total electronic energy in the system with MPE indicates that PMA is in favor.

To independently show the effect of strong interfacial exchange coupling in Bi$_2$Se$_3$/YIG, we have performed electrical transport measurements at low $T$. As shown in Supplementary Fig. 7, we observed a clear negative magnetoresistance (MR) of Bi$_2$Se$_3$/YIG, which is distinct from weak antilocalization (WAL) effect typical of Bi$_2$Se$_3$ films without magnetic perturbation. Detailed analyses show that the MR data can be well reproduced if we assume that the TRS is broken and electrons are magnetically scattered at the bottom surface of Bi$_2$Se$_3$ (see Supplementary Note 4), which may be an indication of the presence of MPE in our Bi$_2$Se$_3$/YIG sample. However, we did not detect anomalous Hall effect in our samples, which might be obscured by the bulk conduction of Bi$_2$Se$_3$ in the transport measurements.

The $H_{eff}$ of the bilayer samples, again, shows different $T$ evolution than that of the bare YIG in Fig. 3f. $H_{eff}$ built up with decreasing $T$ in bilayers while the $H_{eff}$ of the YIG single layer was $T$ independent and close to zero. Phenomenologically, the $H_{eff}$ resembles the exchange bias field of interlayer exchange coupling in an antiferromagnet/ferromagnet interface. However, we would like to exclude the possibility of exchange bias for the following two reasons. First, as shown in Supplementary Fig. 6, we did not observe shifts of magnetization hysteresis loop which is characteristic of an exchange bias effect[43]. Secondly, extending the field sweep to reversed applied field, we found that the FMR spectrum was symmetric with respect to the zero applied field, indicating that the direction of $H_{eff}$ followed that of **M**. The observation is distinct from the magnetization pinning of exchange bias, in which the $H_{eff}$ direction is fixed depending on the interfacial magnetic structure. The fact that $H_{eff}$ existed only in FMR measurement suggests that it comes from spin-pumping-induced spin imbalance at the interface as previously reported[44]. Through exchange coupling to the magnetic layer, the non-equilibrium spin density $\langle\mathbf{S}\rangle_{neq}$ of the TSS gives rise to field-like torque:

$$\mathbf{T}_{FL} = \Delta_{ex}\mathbf{M} \times \langle\mathbf{S}\rangle_{neq}, \qquad (4)$$

where $\Delta_{ex}$ is the exchange coupling constant[45]. ST-FMR experiments on NiFe/Bi$_2$Se$_3$[9] and CoFeB/Bi$_2$Se$_3$[46] showed large

$\mathbf{T}_{FL}$ comparable to the damp-like torque owing to spin–momentum locking of TSS. Since spin pumping is the reciprocal process of ST-FMR, one can expect that the $\mathbf{T}_{FL}$ appears as an exchange effective field in spin pumping. Moreover, we noticed that the $T$ dependence of $H_{eff}$ in Fig. 3f resembles that of $\mathbf{T}_{FL}$ in CoFeB/Bi$_2$Se$_3$[46], which implies that $H_{eff}$ and $\mathbf{T}_{FL}$ share the same origin. Although a large $\mathbf{T}_{FL}$ can originate from other systems with strong SOC such as Rashba-split quantum well state[45], which is likely to coexist with the TSS in Bi$_2$Se$_3$/YIG[47], the $\mathbf{T}_{FL}$ from Rashba state is expected to decrease with decreasing Rashba coefficient at low $T$[48]. Here, we highlight that $H_{eff}$ monotonically increased at low $T$. The unique $T$ dependence of $H_{eff}$ suggests that it is likely to originate from TSS.

**Zero-field FMR of Bi$_2$Se$_3$/YIG.** Finally, we demonstrated that the TSS-modulated magnetic anisotropy and $H_{eff}$ in Bi$_2$Se$_3$/YIG are strong enough to induce FMR without an applied field $H_{ext}$, which we term zero-field FMR. Figure 4a displays $T$ evolution of FMR first derivative spectra of Bi$_2$Se$_3$(25)/YIG(15) at $f = 3.5$ GHz. The spectral shape started to deform when the $H_{res}$ was approaching zero. The sudden twists at $H_{ext} \sim +30$ ($-30$) for positive (negative) field sweep arose from magnetization switching of YIG, and therefore led to hysteric spectra. The two spectra merged at 25 K and then separated again when $T$ was further decreased. Figure 4b shows the microwave absorption intensity $I$ spectra with positive field sweeps. We traced the peak position of $I$ spectrum $H_{peak}$ using the red dashed line, and found it coincided with zero $H_{ext}$ at the zero-field FMR temperature $T_0 \sim 25$ K. Below 25 K, $H_{peak}$ moved across the origin and one needed to reverse $H_{ext}$ to counter the internal effective field comprised of the demagnetization field $4\pi M_s$, $H_{int}$, and $H_{eff}$ (Fig. 4e). It should be pointed out that the presence of $H_{int}$ alone would be inadequate to realize zero-field FMR. Only when $H_{eff}$ is finite would the system exhibit intercepts as we have seen in Fig. 3b. We further calculate $T_0$ as a function of microwave excitation frequency $f$ (Fig. 4f) using Eq. (2) and the extracted $H_{eff}$ of Fig. 3f. We obtain that, with finite $H_{eff}$ persisting up to room temperature, zero-field FMR can be realized at high $T$ provided $f$ is sufficiently low. However, we emphasize that it is advantageous for YIG to be microwave-excited above 3 GHz. When $f < 3$ GHz, parasitic effects such as three-magnon splitting[49,50] take place and significantly decrease the microwave absorption in YIG. Here, we demonstrate that the strong exchange coupling between Bi$_2$Se$_3$ and YIG gave rise to zero-field FMR in the feasible high frequency

operation regime of YIG. Further improvement of interface quality of $Bi_2Se_3$/YIG is expected to raise $H_{eff}$ and $T_0$ for room-temperature, field free spintronic application.

## Discussion

Most experiments probing the spin transfer or spin–charge interconversion at TSS used FMs as the spin source/detector. The pitfall of FMs is that the constituent transition metals are chemically reactive with chalcogenides. Severe reactions can occur when an FM is deposited on a TI, forming new species that complicated the system under study. Even if an ideal TI/FM interface is achieved, theoretical study suggests that the electron doping from the FM can significantly shift the Fermi level of TIs and destroy the spin texture[27]. Besides, for SOT generation, current-shunting by FM reduces the current flowing in the TI and diminishes the SOT strength. Therefore, ferromagnetic insulators such as YIG is a far better platform to study the coupling mechanism between TSS and magnetic layers.

We attribute the high-temperature interfacial IMA to the enhanced exchange coupling of $Fe^{3+}$ ions in YIG mediated by TSS based on the $d_{BS}$ dependence of $K_i$ in Fig. 2c. We emphasize that, although the model in ref. [42] predicts a PMA originated from direct exchange coupling between TSS and a magnetic layer, in reality, other contributions of magnetic anisotropy dependent on the detailed interfacial atomic structure can arise. As illustrated in ref. [31], in addition to the PMA from MPE, the stress anisotropy energy of EuS can also be magnified by the strong SOC of $Bi_2Se_3$, which would not necessarily be PMA for a material system other than EuS/$Bi_2Se_3$. Other factors such as the Fermi energy of $Bi_2Se_3$ can have pronounced effects on the exchange coupling constant and total anisotropy energy[31]. Given the multiple sources of magnetic anisotropy that are possibly influenced by TSS, an in-depth theoretical study will be needed to precisely describe the high-temperature interfacial IMA and the emerging low-temperature PMA of $Bi_2Se_3$/YIG.

The TSS-modulated magnetization dynamics presented in this work have important implications. Firstly, the electronic structure of TI/ferromagnetic insulator interface has a pronounced influence on the magnetization dynamics. It should be noted that the strong coupling between the TSS and YIG can potentially modify the TSS of pure $Bi_2Se_3$. Since the spin texture of TSS is critical for spin transport, an insertion layer may be needed to decouple YIG and $Bi_2Se_3$ for spintronics devices. The interface structure, in turn, depends strongly on the sample fabrication process. For example, Wang et al.[8] reported a markedly different $d_{BS}$ dependence of $g_{\uparrow\downarrow}$ from the one shown in Fig. 2d. Specifically, our samples show larger $\Delta\tilde{\alpha}$ when $d_{BS}$ was approaching the 2D limit. Note that the linewidth broadening observed in this work is overall larger than that reported in ref. [8] mainly because we have chosen thinner YIG films. The discrepancy in the $d_{BS}$ dependence of $g_{\uparrow\downarrow}$ may be reconciled by the different sample characteristics by comparing the TEM images and the surface morphology of $Bi_2Se_3$, etc.

Secondly, although the interface spin structure is of great interest to investigate, it has been difficult to measure with spin-polarized photoemission techniques because of the limited probing depth. Spin pumping provides another route to resolve the problem, since it has proven to be a powerful tool to probe magnetic phase transition of ultrathin films[40,41]. Here, we extended the concept and used spin pumping to study the MPE in $Bi_2Se_3$/YIG. The indicators of MPE are shown in Fig. 3c, d. Further testing of the validity of this method will depend on the improvements in the sample quality, such as a sharper interface and lowering the carrier density of $Bi_2Se_3$.

Lastly, the observations of large $K_i$, $\Delta\tilde{\alpha}$ at room temperature, and $H_{eff}$ at low temperature in $Bi_2Se_3$/YIG echo the theoretical

predictions of the magnetization dynamics of a perpendicularly magnetized layer interacting with TSS[51–53]. According to these models, the gap opening of TSS due to broken TRS leads to topological (inverse) spin galvanic effect[51,52], anisotropic shifts of FMR frequency[52], and anisotropic damping[53]. Despite the fact that an interfacial PMA showed up at low temperature in $Bi_2Se_3$/YIG, the bilayer sample still exhibited a gross in-plane anisotropy due to the shape anisotropy of YIG. However, the notable modulation of the YIG properties presented in this work is a promising start to examine these models. We expect ferromagnetic insulators with PMA, such as strained TmIG[54], will offer new opportunities to realize the phenomena.

In summary, we have investigated the magnetization dynamics of YIG in the presence of interfacial exchange coupling and TSS of $Bi_2Se_3$. The significantly modulated magnetization dynamics at room temperature are shown to be TSS-originated through the $Bi_2Se_3$ thickness dependence study. The temperature-dependent study reveals a possible signature of MPE and an emerging PMA that compensates the high-temperature IMA, with a spin-pumping-induced effective field increasing toward low temperature. The underlying mechanism of these phenomena calls for further theoretical modeling and understanding. To our knowledge, this is the first work that links the magnetization dynamics of the magnetic layer to TSS, showing that FMR and spin pumping can be effective techniques to probe the interface magnetic properties. Moreover, the TSS-modulated dynamics are a cornerstone for future investigation on novel physics such as topological inverse spin galvanic effect, and further raise several interesting topics. For example, how the $H_{eff}$, a quantity that comes from the non-equilibrium process of spin pumping, depends on the spin texture of TSS and the interfacial magnetic anisotropy will be an important question to answer. Temperature-dependent FMR with out-of-plane setup should provide us with valuable information. Therefore, understanding the interplay between these phenomena and further manipulating them will be a step forward toward developing TI-based spintronics.

## Methods

**Sample preparation and structural properties.** The YIG thin films were deposited on (111)-oriented gadolinium gallium garnet (GGG) substrates by off-axis sputtering at room temperature. The GGG(111) substrates were first ultra-sonically cleaned in order of acetone, ethanol, and DI-water before being mounted in a sputtering chamber with the base pressure of $2 \times 10^{-7}$ Torr. For YIG deposition, a 2-inch YIG target was sputtered with the following conditions: an applied rf power of 75 W, an Ar pressure of 50 mtorr, and a growth rate of 0.6 nm/min. The samples were then annealed at 800 °C with an $O_2$ pressure of 11.5 mtorr for 3 h. Supplementary Fig. 1(a) displays the atomic force microscopy (AFM) image of the YIG surface, showing a flat surface with a roughness of 0.19 nm. Supplementary Fig. 1(b) shows the high-angle annular dark-field (HAADF) image of YIG/GGG. The YIG thin film was epitaxially grown on the GGG substrate with excellent crystallinity. No crystal defects were observed at the YIG bulk and YIG/GGG interface.

The YIG/GGG samples were annealed at 450 °C in the MBE growth chamber for 30 min prior to $Bi_2Se_3$ growth at 280 °C. The base pressure of the system was kept about $2 \times 10^{-10}$ Torr. Elemental Bi (7N) and Se (7N) were evaporated from regular effusion cells[55]. As shown in Supplementary Fig. 3(a), streaky reflection high-energy electron diffraction (RHEED) patterns of $Bi_2Se_3$ were observed. Supplementary Fig. 3(b) displays the surface morphology of 7 quintuple layer (QL) $Bi_2Se_3$ taken by AFM. The image shows layer-by-layer growth of $Bi_2Se_3$ with the step heights ~1 nm, which corresponds to the thickness of 1 QL. The surface roughness of our 7 QL $Bi_2Se_3$ is ~0.28 nm within a layer. The layer structure of $Bi_2Se_3$ was also revealed by the HAADF image shown in Supplementary Fig. 3(c). Despite the high-quality growth of $Bi_2Se_3$, an amorphous interfacial layer of ~1 nm formed. The excellent crystallinity of our samples was verified by clear Pendellösung fringes of the synchrotron radiation x-ray diffraction (SR-XRD) data shown in Supplementary Fig. 3(d). The fringes of YIG(444) peak do not show clear changes before and after the growth of $Bi_2Se_3$, indicating that the lattice parameter in the normal direction of YIG remains unchanged. To check the lattice parameter of in-plane direction, we also performed in-plane radial scans of YIG/GGG(22–4) peaks. Supplementary Fig. 3(e) shows that the peaks position measured before and after growing $Bi_2Se_3$ is perfectly matched, indicating the absence of $Bi_2Se_3$-induced

strains in YIG that might contribute additional magnetic anisotropy[56]. Supplementary Fig. 3(f) displays the x-ray reflectivity (XRR) data of our $Bi_2Se_3(6)$/YIG(50) sample. From the fit to the data, we extract the $Bi_2Se_3$ surface roughness and $Bi_2Se_3$/YIG interface roughness to be 0.16 nm and 0.22 nm, respectively. Note that the interface roughness of $Bi_2Se_3$/YIG is close to that of YIG surface (0.19 nm), which means the interdiffusion at the interface is at minimal, if any.

**FMR measurement setup**. To investigate the magnetic properties of $Bi_2Se_3$/YIG, room-temperature angle- and frequency-dependent FMR measurements were performed independently using a cavity and co-planar waveguide, respectively (Fig. 1a, b). For the temperature-dependent FMR, the co-planar waveguide was mounted in a cryogenic probe station (Lake Shore, CPX-HF), which enables samples to be cooled as low as 5 K. The external field is modulated for lock-in detection in all of the measurements. The modulation amplitude was kept below 1/4 of the FMR linewidth to avoid serious spectral distortions. The microwave source power was no larger than 5 dBm.

**Data availability**. The experimental data of this work are available from the corresponding authors upon reasonable request.

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

## Acknowledgements

We would like to thank Prof. Mingzhong Wu, Dr. Hsin Lin, and Dr. Tao Liu for their helpful discussion. We would also like to thank Dr. Jauyn Grace Lin for her technical support and Dr. Chien-Ting Wu for the TEM analyses. The work is supported by MoST 105-2112-M-007-014-MY3, 106-2112-M-002-010, 106-2622-8-002-001, and 105-2112-M-001-031-MY3 of the Ministry of Science and Technology in Taiwan.

## Author contributions

Y.T.F. designed the experiment, collected the FMR data, and analyzed the data. K.H.M.C., C.C.T., C.C.C., and C.N.W. fabricated the samples. C.K.C. performed the XRD measurements and S.R.Y. performed the transport measurements. S.F.L. provided scientific supports. J.K. and M.H. supervised the project. Y.T.F. and J.K. wrote the manuscript with the comments of all the authors.

## Additional information

**Competing interests:** The authors declare no competing financial interests.

