## [Peer Review File · Nature Communications]

Reviewer #1 (Remarks to the Author):

This manuscript reports a study of interfacial coupling in Bi₂Se₃/YIG bilayers with the observation of large in-plane interfacial magnetic anisotropy and FMR damping enhancement. The interfacial anisotropy and damping enhancement show an obvious dependence on the TI film thickness, with the maximum at 6-QL thick Bi₂Se₃, which is attributed to the threshold for the coupling of the two topological surface states. The authors also observe signatures of magnetic proximity effect up to 180 K in the TI/YIG bilayers, and zero-applied-field FMR. There is clear evidence that a strong in-plane magnetic anisotropy emerges in the TI/YIG bilayers as compared to the single YIG films. It is very likely that the induced in-plane anisotropy is due to the interface between Bi₂Se₃ and YIG. This reviewer believes that this manuscript could be accepted for publication in Nature Communications if the authors can address the following questions:

1. To confirm that the induced in-plane magnetic anisotropy and damping enhancement are due to interfacial exchange coupling, not other effects such as inter-diffusion at the interface, more XRD results should be presented. For example, a comparison of the Laue oscillations of the YIG (444) peak and small-angle x-ray reflectivity scans between a single YIG film and after a Bi₂Se₃ layer deposited on that same YIG film will tell valuable information about the TI/YIG interface. Also, what is the peak in the inset of Supplementary Fig. 3d? what is the shoulder? The XRD for a single YIG film should also be provided (in the Supplementary Materials).
2. Supplementary Fig. 6 shows SQUID hysteresis loops for the bilayers, from which the authors should be able to extract saturation magnetization. Then combining with the effective magnetization obtained from FMR measurements, they should be able to get the anisotropy. This is an independent method to measurement anisotropy and will strengthen the argument made in this manuscript.
3. The authors attribute the kinks in Fig. 3c and 3d to the T_c of magnetic proximity effect. This is only a possible reason for the behavior and it should not be taken as a sure conclusion.
4. Some experimental parameters, such as the microwave power for FMR measurements, are not given.

Reviewer #2 (Remarks to the Author):

In this work the authors report an extensive ferromagnetic resonance (FMR) study on Bi₂Se₃ films of varying thickness grown on the magnetic insulator, YIG. The authors compare their measurements to normal YIG films and find enhanced damping like effects as well as a new in-plane magnetic anisotropy at room temperature. They also report some potential indirect evidence of perpendicular induced proximity magnetism at low temperatures. Overall, I find the experimental FMR characterization as being very complete. The authors have measured frequency dependence, angular dependence, and temperature dependence. They also were very thorough in studying films of varying thickness. The three physical effects examined in this work are: spin pumping effects, low temperature perpendicular magnetic anisotropy (PMA) effects, and the (to my knowledge) more novel high temperature in-plane magnetic anisotropy (IMA). The latter effect has special emphasis in the text.

Spin pumping effects (ref. 7) and PMA effects (ref.14, 41) have already been reported in TI/YIG bilayers, and although the dataset that the authors report here is very useful for the community it is probably better suited for a more specialized audience. It is my opinion that the potential impact of this work hinges on the newly reported IMA effect. I find this effect interesting but its origin goes largely unexplained and I think that this overall hurts the paper. The authors do rule out strain effects as being a potential culprit for IMA, and the thickness dependent study does suggest that the effect is interfacial in origin. Still, this is not strong enough to leave topological surface

state physics as the only plausible explanation left for the IMA.

I want to re-emphasize to the authors that I believe the experimental work to be of good quality and believe that the data presented overall is relevant to a specialized audience. However, I cannot recommend publication in Nature Communications because I am not overall convinced that the main novelty of this work (IMA) actually originates from a topological surface state. By way of analogy, theoretical support similar to how ref. 41 supported the direct experimental observation of perpendicular magnetism a TI (ref 14) would greatly benefit this work. A more clear mechanism would be especially useful in this work because the magnetization dynamics in YIG are more of an indirect probe. I do believe that the paper in the current form is in good shape and is close to being suitable for publication, but my opinion is that the authors should consider a more targeted journal.

In order to improve the manuscript for any future (re)submissions I would also suggest that the authors address the following list of concerns.

1) To my eye, the linewidth broadening effects reported in this work are appreciably larger than in spin pumping studies (ref.7) and ST-FMR studies (ref.8). The latter study involved a ferromagnetic metal as opposed to the insulator YIG. Can the authors comment on differences here between studies?

2) At low temperatures the analysis suggests that in the YIG/TI samples an in-plane effective field is present leading to even potentially a zero-applied field resonance. This is another effect that has a somewhat unclear origin as written. As the authors discuss, perpendicular magnetic anisotropy effects are expected to emerge from proximity induced magnetism at low temperature. It is unclear how the in-plane field, phenomenologically added into the Kittel equation, is related to this effect. In fact, the authors have a statement in line 330 that seems to be linking a perpendicularly magnetized layer to the in-plane effective field. I think the authors need to clear this confusing item up in any revision.

3) Related to the second point: Have the authors performed any out-of-plane angular FMR measurements at low temperature to compare with RT data as seen in Figure 1? I imagine that this type of measurement could be beneficial as a way to explore the phenomenology.

More Minor Comments:

Unless I missed it the acronym, TSS, is not defined anywhere in the manuscript. The authors should define this (I think) as topological surface state.

In line 173 the authors state that they are plotting the spin mixing conductance but they are plotting a damping parameter which the mixing conductance is proportional to.

In line 193 the authors state that the data "exhibited negative intercepts at H_{res} ". The data plot is clear but I am not sure what the authors are intending to say with this sentence. Clarification on what the negative intercepts are is needed.

The authors state in line 211 that they are unable to detect lineshapes beyond 100 Oe due to instrumental limits. I'm not sure if this is what they intended to say as they do have field sweeps shown in Figure 1 that presumably would allow for detection of a lineshape of 100 Oe.

Reply to reviewer #1 of Manuscript-17-21884

We would like to thank the reviewers very much for the pertinent comments and important suggestions that have helped improve the content and the quality of our paper a great deal. Here we reply to each question and comment, point by point, in the following:

- 1) "To confirm that the induced in-plane magnetic anisotropy and damping enhancement are due to interfacial exchange coupling, not other effects such as inter-diffusion at the interface, more XRD results should be presented. For example, a comparison of the Laue oscillations of the YIG (444) peak and small-angle x-ray reflectivity scans between a single YIG film and after a Bi_2Se_3 layer deposited on that same YIG film will tell valuable information about the TI/YIG interface. Also, what is the peak in the inset of Supplementary Fig. 3d? what is the shoulder? The XRD for a single YIG film should also be provided (in the Supplementary Materials)."

Our reply

Yes, we agree with the reviewer. To demonstrate that the induced in-plane magnetic anisotropy and damping enhancement are not due to the other effects such as strain or inter-diffusion at the interface, we have modified the presentation of our XRD results more clearly.

Specifically: (1) We have revised the Supplementary Fig. 3(d) to show the normal scan of $\text{Bi}_2\text{Se}_3/\text{YIG}$ and YIG. We can see the Laue fringes of the two samples match each other, indicating the lattice constant c do not clearly change after the growth of Bi_2Se_3 . (2) The inset of the previous manuscript is the in-plane radial scan. We have moved the figure to Supplementary Fig. 3(e) in this revision. The peak is YIG/GGG(22-4), where the main peak is attributed to GGG and the shoulder to YIG. (3) The slightly different peak positions of the YIG and GGG may be due to the lattice relaxation of YIG film in reaching certain thickness. However, the main point here is that data taken before and after the growth of Bi_2Se_3 overlap very well, which means Bi_2Se_3 does not cause notable strain in the YIG film. (4) We have also added the XRR data of our $\text{Bi}_2\text{Se}_3(6)/\text{YIG}(50)$ as Supplementary Fig. 3(f) and extract the surface and interface roughness of $\text{Bi}_2\text{Se}_3/\text{YIG}$. The interface roughness is 0.22 nm close to the surface roughness 0.19 nm of pure YIG film measured by AFM (Supplementary Fig. 1(a)), indicating that the interdiffusion is small. We have revised the Method section accordingly, starting from line 430:

"The fringes of YIG(444) peak does not show clear changes before and after the growth of Bi_2Se_3 , indicating that the lattice parameter in the normal direction of YIG remain unchanged. To check the lattice parameter of in-plane direction, we also performed in-plane radial scans of YIG/GGG(22-4) peaks. Supplementary Fig. 3(e) shows that the peaks position measured before and after growing Bi_2Se_3 are perfectly matched, indicating the absence of Bi_2Se_3 -induced strains in YIG that might contribute additional magnetic anisotropy⁵⁶. Supplementary Fig. 3(f) display the x-ray reflectivity (XRR) data of our $\text{Bi}_2\text{Se}_3(6)/\text{YIG}(50)$ sample. From the fit to the data, we extract the Bi_2Se_3 surface roughness and $\text{Bi}_2\text{Se}_3/\text{YIG}$ interface roughness of 0.16 nm and 0.22 nm, respectively. Note that the interface roughness of $\text{Bi}_2\text{Se}_3/\text{YIG}$ is close to that of YIG surface (0.19 nm), which means the interdiffusion at the interface is at minimal, if any."

, and the figure caption of Supplementary Fig. 3:

"(d) SR-XRD of our Bi₂Se₃(25)/YIG(12) sample. Clear Pendellösung fringes of YIG and Bi₂Se₃ indicates excellent crystallinity. (e) In-plane radial scan of YIG(12) and Bi₂Se₃(25)/YIG(12). The peak at 23.44° is attributed to GGG(22-4), whereas the shoulder to YIG(22-4). (f) XRR results of Bi₂Se₃(6)/YIG(50) and the fit for the extraction of surface and interface roughness."

- 2) "Supplementary Fig. 6 shows SQUID hysteresis loops for the bilayers, from which the authors should be able to extract saturation magnetization. Then combining with the effective magnetization obtained from FMR measurements, they should be able to get the anisotropy. This is an independent method to measurement anisotropy and will strengthen the argument made in this manuscript."

Our reply

We thank the reviewer for an excellent suggestion. In principle, the method suggested by the reviewer should allow us to calculate the effective anisotropy constant K_i using $K_i \approx (1/2)(4\pi M_{eff}^{YIG} - 4\pi M_{eff}^{BS/YIG})M_s d_{YIG}$. However, the major difficulty came from the large paramagnetic signals from the GGG substrates that prevented us from accurately determining the M_s of our samples. Here we present the raw data of our SQUID measurement in the inset of Supplementary Fig. 6. We estimate the errors to be as large as 10 % and conclude that it is not accurate enough for us to further extract K_i reliably. Instead, we can calculate the interfacial anisotropy field $H_{int} \approx 4\pi M_{eff}^{BS/YIG} - 4\pi M_{eff}^{YIG}$ without involving M_s . The temperature dependence of K_i and H_{int} should qualitatively agree as we expect the M_s monotonically increases at low T by no more than 40 % of the RT value.

Hence we added the new data of H_{int} and show its T dependence in the inset of Fig. 3(e). We can see a turning of the H_{int} curves for both samples at a temperature range of 150-180 K, that was attributed to the interfacial PMA resulting from the magnetic proximity effect (MPE) in Bi₂Se₃/YIG. We interpret this as the emergence of an interfacial perpendicular magnetic anisotropy (PMA) that competes with the *in-plane* magnetic anisotropy (IMA). In addition, to independently show the effect of interfacial exchange coupling, we have appended our low T magnetoresistance (MR) data to strengthen our claim (please refer to Supplementary Fig. 7 and Note 4).

Moreover, we have added a section to describe and interpret the H_{int} data starting from line 248 as follows,

"We further calculate the interfacial anisotropy field H_{int} using $4\pi M_{eff}^{BS/YIG} - 4\pi M_{eff}^{YIG} \approx -H_{int}$. The inset of Fig. 3(e) shows the T dependence of H_{int} . The magnitude of H_{int} increased as the samples cooled down from room temperature at first. Upon crossing the "hump" temperature regions, H_{int} magnitude started to decrease with further decreasing T . Although the samples exhibit interfacial IMA ($H_{int} < 0$) within the temperature range of our measurement, further extending the trend of Bi₂Se₃(16)/YIG(17), specifically, leads to interfacial *perpendicular* magnetic anisotropy (PMA) ($H_{int} > 0$) below 40 K. The turning of H_{int} curves around 150 K implied that a competing magnetic anisotropy was emerging, which favored perpendicular direction and effectively diminished the IMA that persisted up to room temperature. Observing that the turning of H_{int} curves were in vicinity of the

individual hump temperature, we thus attribute the interfacial PMA to MPE in Bi₂Se₃/YIG. Our scenario is further supported by a theoretical model that considers direct the exchange coupling of TSS and an adjacent magnetic layer^{31,42}. In this model, the calculated total electronic energy in the system with MPE indicates that perpendicular anisotropy is in favor.

To independently show the effect of strong interfacial exchange coupling in Bi₂Se₃/YIG, we performed electrical transport measurements at low T . As shown in Supplementary Fig. 7, we observed a clear negative magnetoresistance (MR) of Bi₂Se₃/YIG, which is distinct from weak antilocalization (WAL) effect typical of Bi₂Se₃ films without magnetic perturbation. Detailed analyses show that the MR data can be well-reproduced if we assume that the TRS is broken and electrons are magnetically scattered at the bottom Bi₂Se₃ surface (See Supplementary Note 4), which may be indication for the presence of MPE in our Bi₂Se₃/YIG sample. However, we did not detect anomalous Hall effect in our samples, which might be obscured by the bulk conduction of Bi₂Se₃ in the transport measurements."

- 3) "The authors attribute the kinks in Fig. 3c and 3d to the T_c of magnetic proximity effect. This is only a possible reason for the behavior and it should not be taken as a sure conclusion."

Our reply

We agree with the reviewer that the kinks are the possible reason of MPE. We have tuned down our statement in the conclusion by rephrasing the feature to be "possible signatures of MPE".

- 4) "Some experimental parameters, such as the microwave power for FMR measurements, are not given."

Our reply

The microwave power of our microwave source was set to be no larger than 5 dBm. We have added a sentence in the Method section:

"The microwave source power was no larger than 5 dBm."

Reply to reviewer #2 of Manuscript-17-21884

We would like to thank the reviewers very much for the pertinent comments and insightful suggestions that have helped improve the content and the quality of our paper a great deal. Before formally answering the questions of reviewer #2, we would like to respond first to his/her concerns of the origin of the interfacial IMA, which was stated in the preface of the response letter:

"I want to re-emphasize to the authors that I believe the experimental work to be of good quality and believe that the data presented overall is relevant to a specialized audience. However, I cannot recommend publication in Nature Communications because I am not overall convinced that the main novelty of this work (IMA) actually originates from a topological surface state. By way of analogy, theoretical support similar to how ref. 41 supported the direct experimental observation of perpendicular magnetism a TI (ref. 14) would greatly benefit this work. A more clear mechanism would be especially useful in this work because the magnetization dynamics in YIG are more of an indirect probe."

Our reply

We understand the concerns of the reviewer that it seems to be lack of convincing evidence in our work to link the IMA to topological surface state (TSS), after the observation of MPE-induced PMA in EuS/Bi₂Se₃ with several theoretical work in support of the PMA phenomenon. Thanks to the reviewer's suggestion, we are inspired to further look into the possible mechanism of the IMA in our Bi₂Se₃/YIG. We noticed a recent theoretical paper (Kim *et al.* Phys. Rev. Lett. **119**, 027201 (2017), cited as ref. 31) that provides valuable information on how TSS can potentially modulate the magnetic anisotropy of the adjacent layer. Although the DFT calculation results presented in this paper was based on EuS/Bi₂Se₃, we believe the physical concepts are universal to apply to Bi₂Se₃/YIG.

Firstly, we would like to point out that the model presented in ref. 41 (now ref. 42 in the revised manuscript) consider the *direct* exchange coupling between TSS and the magnetic layer, which correctly predicts the PMA in EuS/Bi₂Se₃. However, this is a universal but somewhat over-simplified model. As shown in Fig. 1 of ref. 31, the strong spin-orbit coupling (SOC) of Bi₂Se₃ can magnify the stress anisotropy energy inherent in an EuS thin film. Although it happens that the stress anisotropy is also PMA for the optimized structure of EuS/Bi₂Se₃, the calculation shows that such Bi₂Se₃-enhanced anisotropy can exhibit IMA instead, if we vary the EuS lattice parameter. The results imply that, in addition to the PMA from direct exchange, there exists other source of magnetic anisotropy that may be too intricate to describe with a simple model. The contribution to the total magnetic anisotropy should depend on actual interfacial atomic structure, which is considerably different for other materials such as Bi₂Se₃/YIG. However, we think the underlying physics mechanism is clear: the TSS of Bi₂Se₃ significantly enhance the magnetic anisotropy by mediating the exchange coupling between the magnetic ions. Turning to our work, a pronouncedly enhanced magnetic anisotropy of Bi₂Se₃/YIG is one of the major findings in this work. Therefore, together with the unique d_{BS} dependence of K_i shown in Fig. 2(c), the interfacial IMA is inferred to be a plausible consequence of TSS modulation. Hence, we revised the relevant paragraph starting from line 131 as follows:

"The sizable interfacial IMA can be expected given the large SOC of Bi₂Se₃. One possible mechanism is that the electrons at the interface re-distribute upon the hybridization between the Fe *d*-orbital of YIG and the Dirac surface state of Bi₂Se₃. Recent theoretical study on EuS/Bi₂Se₃ bilayers indicate that in addition to the strong SOC, TSS play a crucial role in mediating the exchange coupling of the ions in the magnetic layer³¹. The hybridization between TSS and the magnetic layer can overall enhance the magnetic anisotropy

energy that is inherent at the interface³¹."

, and added a paragraph in the Discussion section (line 340):

"We attribute the high temperature interfacial IMA to the enhanced exchange coupling of Fe³⁺ ions in YIG mediated by TSS based on the d_{BS} dependence of K_i in Fig. 2(c). We emphasize that, although the model in ref. 42 predicts a PMA originated from direct exchange coupling between TSS and a magnetic layer, in reality, other contributions of magnetic anisotropy dependent on the detailed interfacial atomic structure can arise. As illustrated in ref. 31, in addition to the PMA from MPE, the stress anisotropy energy of EuS can also be magnified by the strong SOC of Bi₂Se₃, which would not necessarily be PMA for a material system other than EuS/Bi₂Se₃. Other factors such as the Fermi energy of Bi₂Se₃ can have pronounced effects on the exchange coupling constant and total anisotropy energy³¹. Given the multiple sources of magnetic anisotropy that are possibly influenced by TSS, an in-depth theoretical study will be needed to precisely describe the high T interfacial IMA and the emerging low T PMA of Bi₂Se₃/YIG." (the low T PMA will be discussed below)

Indeed, a precise description of the interfacial magnetic anisotropy of Bi₂Se₃/YIG calls for further theoretical understanding, which would probably invoke first-principle calculations. However, the complicated crystal structure of YIG has long been hampering the progress. Therefore, our experimental results provide exceptionally valuable information on this important yet difficult-to-model TI/YIG system.

Secondly, as requested by reviewer #1, we have calculated the interfacial anisotropy field H_{int} for various T . The result is shown in the inset of Fig. 3(e). Clearly, there exhibits a peak around 150 K for both samples. We interpret the feature as an emerging low T PMA that competes with the high T IMA. The interpretation is simultaneously based on the fact that temperature of the H_{int} peak position is right below that of the "hump" position, which we has identified as possible signatures of MPE. The observations do echo the direct exchange model in ref. 42 and the latter part of ref. 31. In addition, we have appended the magnetoresistance (MR) data of Bi₂Se₃/YIG to independently show the negative MR effects in support of MPE. Accordingly, we have revised the manuscript starting from line 248:

"We further calculate the interfacial anisotropy field H_{int} using $4\pi M_{eff}^{BS/YIG} - 4\pi M_{eff}^{YIG} \approx -H_{int}$. The inset of Fig. 3(e) shows the T dependence of H_{int} . The magnitude of H_{int} increased as the samples cooled down from room temperature at first. Upon crossing the "hump" temperature regions, H_{int} magnitude started to decrease with further decreasing T . Although the samples exhibit interfacial IMA ($H_{int} < 0$) within the temperature range of our measurement, further extending the trend of Bi₂Se₃(16)/YIG(17), specifically, leads to interfacial *perpendicular* magnetic anisotropy (PMA) ($H_{int} > 0$) below 40 K. The turning of H_{int} curves around 150 K implied that a competing magnetic anisotropy was emerging, which favored perpendicular direction and effectively diminished the IMA that persisted up to room temperature. Observing that the turning of H_{int} curves were in vicinity of the individual hump temperature, we thus attribute the interfacial PMA to MPE in Bi₂Se₃/YIG. Our scenario is further supported by a theoretical model that considers direct the exchange coupling of TSS and an adjacent magnetic layer^{31,42}. In this model,

the calculated total electronic energy in the system with MPE indicates that perpendicular anisotropy is in favor.

To independently show the effect of strong interfacial exchange coupling in $\text{Bi}_2\text{Se}_3/\text{YIG}$, we performed electrical transport measurements at low T . As shown in Supplementary Fig. 7, we observed a clear negative magnetoresistance (MR) of $\text{Bi}_2\text{Se}_3/\text{YIG}$, which is distinct from weak antilocalization (WAL) effect typical of Bi_2Se_3 films without magnetic perturbation. Detailed analyses show that the MR data can be well-reproduced if we assume that the TRS is broken and electrons are magnetically scattered at the bottom Bi_2Se_3 surface (See Supplementary Note 4), which may be indication for the presence of MPE in our $\text{Bi}_2\text{Se}_3/\text{YIG}$ sample. However, we did not detect anomalous Hall effect in our samples, which might be obscured by the bulk conduction of Bi_2Se_3 in the transport measurements."

For the MR data and analyses, please refer to Supplementary Fig. 7 and Note 4.

Finally, we would like to emphasize the impact of this work to the reviewer. From the technical aspect, we agree that FMR is a somewhat indirect probe, compared to other direct techniques such as polarized neutron reflectivity (PNR). However, allow us to point out that, it would be difficult, if possible, to probe the interfacial magnetism using conventional techniques such as VSM or SQUID magnetometer. As we have shown in the revised Supplementary Fig. 6, the dramatically increased low T paramagnetic signals from the GGG substrates overwhelmed the YIG signals. We estimate the GGG signals to be at least 3 order of magnitude larger than that of YIG at the saturation field below 100 K. The proximity-induced magnetic moment is even harder to detect. It is not easy to circumvent the difficulty because GGG is the only substrate currently known for fabricating high quality YIG film. Hence, FMR or spin pumping is a very suitable table-top technique readily available to many researchers worldwide, and that can reliably probe the interfacial magnetic properties without resorting to large facility such as XMCD or neutron scattering.

Moreover, although a number of work has focused on MPE of TI/YIG, to our knowledge, this is the first work to study the interfacial magnetic anisotropy of TI/YIG. Although MPE-induced PMA in $\text{EuS}/\text{Bi}_2\text{Se}_3$ has been reported¹⁵, the magnetic anisotropy of TI/YIG caused by TSS remains largely unknown for the community. We believe the interfacial magnetic anisotropy of TI/YIG is at least as important as that of $\text{EuS}/\text{Bi}_2\text{Se}_3$ because YIG has even broader applications. It is the technical difficulties that hampers the progress on this topic. Therefore, we chose FMR, a standard technique to accurately determine magnetic anisotropy of materials. To emphasize these points we have re-written the introduction paragraph accordingly from line 54 to 61.

Furthermore, there have been disputes in spin pumping results reported so far with large variations from group to group. With the rapid growth of the exciting field, we believe that our work will benefit all researchers in general, who are interested and eager to know the important magnetic feature such as interfacial magnetic anisotropy of TI/YIG, and its technological implications to viable applications of this quantum materials for spintronics in future. Hence our work is not targeted at a specialized audience only.

Now, we reply to each question and comment of the reviewer, point by point, in the following:

- 1) "To my eye, the linewidth broadening effects reported in this work are appreciably larger than in spin pumping studies (ref.7) and ST-FMR studies (ref.8). The latter study involved a ferromagnetic metal as opposed to the insulator YIG. Can the authors comment on differences here between studies?"

Our reply

The linewidth broadening of our samples is overall larger mainly because we intentionally choose thinner YIG films to magnify the H_{int} and linewidth broadening, based on macrospin model. This would ensure adequately larger changes of ΔH and H_{res} for more accurate determination of the damping enhancement $\Delta\tilde{\alpha}$ and $4\pi M_{eff}$. Hence, it would be fairer to compare the damping enhancement $\Delta\tilde{\alpha}$ or spin mixing conductance, in which the effect of YIG thickness has been incorporated. The main difference of $\Delta\tilde{\alpha}$ between ref. 7 (now ref. 8 of the revised manuscript) and our work is that we observed a large enhancement of damping around the 2D limit of Bi_2Se_3 , while the $\Delta\tilde{\alpha}$ shown in Fig. 4(b) of ref. 7 had weak d_{BS} dependence. As described in line 358, we think that the discrepancy could come from different quality of Bi_2Se_3 thin films. For example, the surface of our 7 nm Bi_2Se_3 shows step-like feature that indicates layer-by-layer growth (roughness ~ 0.28 nm), which is shown in Supplementary Fig. 3(b). For comparison, the surface of 6 nm Bi_2Se_3 shown in Fig. 1(b) of ref. 7 doesn't clearly exhibit the step-like feature (roughness ~ 0.71 nm), and pinholes spreading throughout the image can be easily seen. To be more specific in describing the difference of $\Delta\tilde{\alpha}$, we added some sentences starting from line 361:

"Specifically, our samples show larger $\Delta\tilde{\alpha}$ when d_{BS} was approaching the 2D limit. Note that the linewidth broadening observed in this work is overall larger than that reported in ref. 8 mainly because we have chosen thinner YIG films."

As for ref. 8 (now ref. 9), we think that it would be inappropriate to directly compare the linewidth broadening or $\Delta\tilde{\alpha}$ of ferromagnetic metal (FM)/TI and TI/FI heterostructures side by side. From the material aspect, the much cleaner interface of TI/FI has minimized the damping that can potentially arise from the interdiffusion or surface roughness common in FM/TI (see Gupta *et al.* AIP Adv. **7**, 055919 (2017), for example). On the other hand, spin backflow in FM has also been considered to be a correction of damping enhancement (Jiao *et al.* Phys. Rev. Lett. **110**, 217602 (2013)). The two side effects are believed to be minor in spin pumping from YIG to TI.

- 2) "At low temperatures the analysis suggests that in the YIG/TI samples an in-plane effective field is present leading to even potentially a zero-applied field resonance. This is another effect that has a somewhat unclear origin as written. As the authors discuss, perpendicular magnetic anisotropy effects are expected to emerge from proximity induced magnetism at low temperature. It is unclear how the in-plane field, phenomenologically added into the Kittel equation, is related to this effect. In fact, the authors have a statement in line 330 (original version) that seems to be linking a perpendicularly magnetized layer to the in-plane effective field. I think the authors need to clear this confusing item up in any revision."

We would like to split this question into three parts (a-c) for easy to discuss step by step.

- a) "At low temperatures the analysis suggests that in the YIG/TI samples an in-plane effective field is present leading to even potentially a zero-applied field resonance. This is another effect that has a somewhat unclear origin as written."

Our reply

We have discussed the origin of the H_{eff} in the manuscript starting from line 276. To describe the origin and physical mechanism of H_{eff} more clearly, we would like to begin with comparing the experimental observations of our spin pumping in $\text{Bi}_2\text{Se}_3/\text{YIG}$ and ST-FMR measurement on $\text{Py}/\text{Bi}_2\text{Se}_3$ in ref. 9. In our experiment, we have observed a shift of resonance field H_{res} at all frequencies shown in Fig. 3(b), and based on the observation we quantify the shifts using the effective field H_{eff} . Similarly, in Extended Fig. 1 of ref. 9, there is also a H_{res} shift, which the authors attributed to the exchange effective field due to the field-like spin torque from the current-induced spin accumulation at TSS. Note that one of the main feature of ref. 9 is the observation of large field-like torque comparable to damp like torque. This is the first clue suggesting that the H_{eff} is related to TSS.

The second clue is revealed by comparing our T dependence of H_{eff} with the T dependence of field-like torque of another ST-FMR measurement reported in ref. 46. The striking similarity can be seen in Fig. 4 of ref. 46, showing that the field-like spin torque substantially *increased* with decreasing T . Note that the T dependence is unique for TI, as pointed out by the authors on page 4 of ref. 46, since the field-like torque from Rashba splitting has been expected to *decrease* with decreasing T . Furthermore, the field-like torque of Ta/CoFeB exhibit similar decreasing behavior. Hence, the unique T dependence of H_{eff} in our work further support H_{eff} is TSS-originated.

In fact, theory has shown that field-like spin torques do arise from interface with strong SOC (for example, see ref. 45 and Haney *et al.*, Phys. Rev. B **87**, 174411 (2013)). Specifically, for TSS and Rashba-split 2DEG, the torque enters the LLG equation as Eq. 4, where the *non-equilibrium* spin density act as an effective field through a cross product with the magnetization. If we consider the fundamental equivalence of spin pumping effect and ST-FMR, then it is natural to view the H_{eff} in our work as the counterpart effect of the field-like torque observed in ST-FMR.

To strengthen our claim, we have cited the following two papers to highlight the T dependence of H_{eff} :

⁴⁷Bahramy, M. S. *et al.* Emergent quantum confinement at topological insulator surfaces. *Nat. Commun.* **3**, 1159 (2012).

⁴⁸Eldridge, P. S. *et al.* All-optical measurement of Rashba coefficient in quantum wells. *Phys. Rev. B* **77**, 125344 (2008).

, and revised the relevant paragraph starting from line 294:

"Moreover, we noticed that the T dependence of H_{eff} in Fig. 3(f) resembles that of \mathbf{T}_{FL} in $\text{CoFeB}/\text{Bi}_2\text{Se}_3$ ⁴⁶, which implies that H_{eff} and \mathbf{T}_{FL} share the same origin. Although a large \mathbf{T}_{FL} can originate from other systems with strong SOC such as Rashba-split quantum well state⁴⁵, which is likely to coexist with the TSS in $\text{Bi}_2\text{Se}_3/\text{YIG}$ ⁴⁷, the \mathbf{T}_{FL} from Rashba state is expected to decrease with decreasing Rashba coefficient at low T ⁴⁸. Here, we highlight that H_{eff} monotonically increased at low T . The unique T dependence of H_{eff} suggests that it is likely originated from TSS."

b) "As the authors discuss, perpendicular magnetic anisotropy effects are expected to emerge from

proximity induced magnetism at low temperature. It is unclear how the in-plane field, phenomenologically added into the Kittel equation, is related to this effect.”

Our reply

Following the train of thought of the previous part, the H_{eff} is a non-equilibrium phenomenon due to spin pumping. As we have discussed in the manuscript, H_{eff} is phenomenologically similar to exchange bias effective field, which is a *static* and *equilibrium* effect, and we have precluded the possibility by showing hysteresis loops without exchange bias (Supplementary Fig. 6). The main point to clarify here is that, the magnetic anisotropy discussed in this work, high T IMA and low T PMA, are *static* and *equilibrium* magnetic properties. In contrast, spin pumping effect involves *dynamical* exchange at interface and further induces a *non-equilibrium* spin density that manifests as H_{eff} . Therefore, the two types of effects may not be treated on equal footings. The two types of effects surely can have some kind of mutual interaction, but at this stage, we tend to emphasize the different nature of these effects and not to describe their interaction as it would require advanced modeling.

- c) “In fact, the authors have a statement in line 330 that seems to be linking a perpendicularly magnetized layer to the in-plane effective field. I think the authors need to clear this confusing item up in any revision.”

Our reply

Actually, we did not intend to link the PMA to H_{eff} . The purpose of this paragraph was to highlight that the strongly modulated magnetic properties can be viewed as good signs for future study on the magnetization dynamics of a PMA films modulated by TIs. Here, we have revised the paragraph to express our point more clearly in line 380:

"Despite the fact that an interfacial PMA showed up at low T in $\text{Bi}_2\text{Se}_3/\text{YIG}$, the bilayer sample still exhibited a gross in-plane anisotropy due to the shape anisotropy of YIG. However, the notable modulation of the YIG properties presented in this work is a promising start to examine these models."

- 3) “Related to the second point: Have the authors performed any out-of-plane angular FMR measurements as low temperature to compare with RT data as seen in Figure 1? I imagine that this type of measurement could be beneficial as a way to explore the phenomenology.”

Our reply

Unfortunately, our equipment does not allow us to perform *out-of-plane* FMR at low T unless we make a major modification. We believe that the measurements will give valuable information to explore the origin of H_{eff} . As for $4\pi M_{eff}$, the *in-plane* frequency-dependent FMR data presented in this paper should provide equivalent information to that provided by angle-dependent FMR (see Supplementary Note 2). To bring up the future work, we revised the conclusion part starting from line 394:

"Moreover, the TSS-modulated dynamics is a cornerstone for future investigation on novel physics such as topological inverse spin galvanic effect, and further raises several interesting topics. For example, how the H_{eff} , a quantity that comes from the non-equilibrium process of spin pumping, depends on the

spin texture of TSS and the interfacial magnetic anisotropy will be an important question to answer. Temperature-dependent FMR with *out-of-plane* setup should provide us with valuable information."

More Minor Comments:

"Unless I missed it the acronym, TSS, is not defined anywhere in the manuscript. The authors should define this (I think) as topological surface state."

Our reply

Yes, TSS stands for topological surface state. We have added the definition of the acronym in line 56.

"In line 173 the authors state that they are plotting the spin mixing conductance but they are plotting a damping parameter which the mixing conductance is proportional to."

Our reply

This is a typo. We have removed the sentence in this revision.

"In line 193 the authors state that the data "exhibited negative intercepts at H_{res} ". The data plot is clear but I am not sure what the authors are intending to say with this sentence. Clarification on what the negative intercepts are is needed."

Our reply

By observing Eq. (2), the magnitude of the negative intercepts equal H_{eff} . We specifically point out the feature to convince readers that it is necessary to add the H_{eff} term in the Kittel equation to fit the data in Fig. 3(b). It is clear that the original Kittel equation without the H_{eff} is unable to produce an intercept. To clarify, we have added a sentence in line 203: "Note that the Kittel equation in its original form cannot produce an intercept."

"The authors state in line 211 that they are unable to detect lineshapes beyond 100 Oe due to instrumental limits. I'm not sure if this is what they intended to say as they do have field sweeps shown in Figure 1 that presumably would allow for detection of a lineshape of 100 Oe."

Our reply

Here we are indicating the limit of our FMR measurement using co-planar waveguide. Fig. 1(c) is the data taken with a microwave cavity, which has much better sensitivity. To avoid the confusion, we have revised the sentence in line 219: "We were not able to detect FMR signals with ΔH beyond 100 Oe due to the limited sensitivity of our co-planar waveguide."

Reviewer #1 (Remarks to the Author):

The authors have addressed my questions and comments, and added new data in the manuscripts as well as in the supplementary materials. I recommend its acceptance by Nature Communications.

Reviewer #2 (Remarks to the Author):

I have received and read the response letter from Fanchiang et al. on "Strongly exchange-coupled and surface-states-modulated magnetization dynamics in Be₂Se₃/YIG heterostructures".

My concern with the original manuscript was centered around the fact that the IMA the authors can extract from FMR measurements was tied to a coupling between the topological surface state and the magnetic insulator. With most of the related literature showing PMA effects in these same systems (from proximity exchange interaction), I did not feel that the authors effectively argued why the IMA was also a possibility. So, while I felt that the experiment was thorough and well executed with interesting data, I believed that the conclusion was not supported and that the conflicts with literature were not addressed.

I appreciate the efforts that the authors went through in their attempt to better contextualize and interpret their results. I think that the authors now have been able to argue to their audience that the PMA seen in a TI/EuS system is not universal and that IMA in the TI/YIG system could be an example of other interfacial induced anisotropies. To support this, the authors have based an argument on recent theoretical work in TI/EuS where indeed, changes in the lattice parameter of the EuS can change PMA to IMA. In their response I think the following two points are well-emphasized:

1. I do agree with the authors that studying the magnetic anisotropy of the TI/YIG "is at least as important" as the TI/EuS system. The authors are correct to note that YIG is a material that is of broad interest in the field of spintronics, magnonics, etc. I do appreciate the fact that this is an FMR study in the first place is largely due to the YIG being an excellent material for this type of study.
2. I think that the authors have successfully argued in their rebuttal that the audience should not be so quick to immediately equivocate the magnetic proximity effect with what they call interfacial magnetic anisotropy. Essentially, the authors remind us that it is in EuS/Bi₂Se₃ where PMA was observed and that "the magnetic anisotropy of TI/YIG...remains largely unknown."

The other concerns and comments I had suggested to the authors in my first review have been addressed and I am satisfied with the changes that were made. I can now recommend publication of this manuscript and feel the authors have done a commendable job improving their original submission.

Reply to reviewer #1 of Manuscript-17-21884A-Z

The authors have addressed my questions and comments, and added new data in the manuscripts as well as in the supplementary materials. I recommend its acceptance by Nature Communications.

Our reply

We thank the referee for his/her positive comments and recommendation of publication in Nature Communications.

Reply to reviewer #2 of Manuscript-17-21884A-Z

I have received and read the response letter from Fanchiang et al. on “Strongly exchange-coupled and surface-states-modulated magnetization dynamics in Be₂Se₃/YIG heterostructures”.

My concern with the original manuscript was centered around the fact that the IMA the authors can extract from FMR measurements was tied to a coupling between the topological surface state and the magnetic insulator. With most of the related literature showing PMA effects in these same systems (from proximity exchange interaction), I did not feel that the authors effectively argued why the IMA was also a possibility. So, while I felt that the experiment was thorough and well executed with interesting data, I believed that the conclusion was not supported and that the conflicts with literature were not addressed.

I appreciate the efforts that the authors went through in their attempt to better contextualize and interpret their results. I think that the authors now have been able to argue to their audience that the PMA seen in a TI/EuS system is not universal and that IMA in the TI/YIG system could be an example of other interfacial induced anisotropies. To support this, the authors have based an argument on recent theoretical work in TI/EuS where indeed, changes in the lattice parameter of the EuS can change PMA to IMA. In their response I think the following two points are well-emphasized:

1. I do agree with the authors that studying the magnetic anisotropy of the TI/YIG “is at least as important” as the TI/EuS system. The authors are correct to note that YIG is a material that is of broad interest in the field of spintronics, magnonics, etc. I do appreciate the fact that this is an FMR study in the first place is largely due to the YIG being an excellent material for this type of study.

2. I think that the authors have successfully argued in their rebuttal that the audience should not be so quick to immediately equivocate the magnetic proximity effect with what they call interfacial magnetic anisotropy. Essentially, the authors remind us that it is in EuS/Bi₂Se₃ where PMA was observed and that “the magnetic anisotropy of TI/YIG...remains largely unknown.”

The other concerns and comments I had suggested to the authors in my first review have been addressed and I am satisfied with the changes that were made. I can now recommend publication of this manuscript and feel the authors have done a

commendable job improving their original submission.

Our reply

We thank the reviewer for his/her appreciation for the interpretation of our results, and his/her recommendation of publication.